# The Influence of Tree Structural and Species Diversity on Temperate Forest Productivity and Stability in Korea

**Juhan Park [1,2], Hyun Seok Kim [1,2,3,4,*] 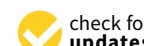, Hyun Kook Jo [5] and II Bin Jung [5]**

[1] National Center for Agro-Meteorology, Seoul 08826, Korea; ecohani@gmail.com
[2] Department of Forest Sciences, Seoul National University, Seoul 08826, Korea
[3] Interdisciplinary Program in Agricultural and Forest Meteorology, Seoul National University, Seoul 08826, Korea
[4] Research Institute for Agriculture and Life Sciences, Seoul National University, Seoul 08826, Korea
[5] Forest Resources Information Division, Korea Forest Promotion Institute, Seoul 07570, Korea; hcho@kofpi.or.kr (H.K.J.); leohunter@kofpi.or.kr (I.B.J.)
* Correspondence: cameroncrazies@snu.ac.kr

**Abstract:** Research Highlights: Using a long-term dataset on temperate forests in South Korea, we established the interrelationships between tree species and structural diversity and forest productivity and stability, and identified a strong, positive effect of structural diversity, rather than tree species diversity, on productivity and stability. Background and Objectives: Globally, species diversity is positively related with forest productivity. However, temperate forests often show a negative or neutral relationship. In those forests, structural diversity, instead of tree species diversity, could control the forest function. Materials and Methods: This study tested the effects of tree species and structural diversity on temperate forest productivity. The basal area increment and relative changes in stand density were used as proxies for forest productivity and stability, respectively. Results: Here we show that structural diversity, but not species diversity, had a significant, positive effect on productivity, whereas species diversity had a negative effect, despite a positive effect on diversity. Structural diversity also promoted fewer changes in stand density between two periods, whereas species diversity showed no such relation. Structurally diverse forests might use resources efficiently through increased canopy complexity due to canopy plasticity. Conclusions: These results indicate reported species diversity effects could be related to structural diversity. They also highlight the importance of managing structurally diverse forests to improve productivity and stability in stand density, which may promote sustainability of forests.

**Keywords:** temperate forest; species diversity; structural diversity; basal area increment; stability; structural equation model

## 1. Introduction

The relationships between productivity and species diversity are one of the oldest questions in ecological studies. For the last two decades, this relationship has been intensively investigated [1–3]. However, considerable variations and controversy in this relationship still remain, and the mechanisms underlying this relationship are not yet fully understood.

The complementarity hypothesis links species diversity and productivity. It assumes that productivity increases with species number through positive interactions such as competition reduction or facilitation among species. By comparison, facilitation occurs when a species improves the growing conditions for another species through processes such as nitrogen fixation [4] and hydraulic

redistribution [5]. Competition reduction occurs when a species is introduced to a stand that has intense intra-specific competition, and replaces it with less intense inter-specific competition though the processes such as stratification of the root system or canopy [6]. However, it is unknown whether the positive effects of species diversity on productivity applies in forests, as the majority of evidence on the relationship has been found on grasslands [1,7].

The evidence for positive effects of tree species diversity on forest productivity has been accumulating [8,9], but negative or neutral effects of tree species diversity on productivity have also been reported [10,11]. Despite the recent findings of the positive relationship between tree species diversity and forest productivity [12,13], there are remaining questions as to why different relationships are observed, especially in temperate forests [8,11]. These might be caused by functional redundancy and overlap of niches among coexisting tree species [14], and different diversity measures, such as functional or phylogenic diversity, have been considered to explain the diversity–productivity relationship. Unlike short-lived ecosystems such as grasslands, forests are composed of long-lived tree species with large inter- and intra-specific size differences. This heterogeneity in tree size might have similar effects to species diversity, as trees of different sizes among the same species can occupy different niches. Therefore, structural diversity is considered as a dominant factor in forest productivity and standing biomass [15,16].

Species and structural diversity can also be inter-correlated, as species diversity increases structural diversity when species with different life strategies coexist. Species diversity can also promote tree size and canopy height heterogeneity as well [17], while structural diversity can be a proxy for species diversity. Thus, species diversity indirectly affects productivity via structural diversity. This could explain the positive relationship between them at a global scale. However, structural diversity could negatively affect productivity in monocultures [18,19], and have opposing effects in mixed stands [16,20]. Therefore, the underlying mechanism of diversity and productivity remains debated.

The species and structural diversity also affects forest stability. However, the effects of diversity on forest ecosystem stability are rarely examined, partly as forest stability can be defined in multiple ways, including resistance, resilience, constancy, and persistence in either or both functional and structural aspects [21], which causes divergence within the field [22]. In addition, the diversity–stability relationship could be connected to the diversity–productivity relationship. Two main mechanisms for the diversity–stability relationship are proposed—asynchrony and over-yielding [23]. Asynchrony, which indicates an imperfect correlation in species responses to stress and disturbance, mitigates the negative effects of disturbance and decreases the overall variation in ecosystems. Over-yielding, which occurs when a mixed species forest has higher productivity than the sum of expected productivity of each species, is facilitated by niche partitioning [24], and increases forest stand resistance against disturbance.

Thus, this study aims to find dominant factors influencing forest productivity and stability in temperate forests. By using National Forest Inventory (NFI) data collected in South Korea at 5-year intervals, we constructed a structural equation model (SEM) to test the direct and indirect effects of species and structural diversity on forest productivity. We hypothesized that: 1. species diversity would have positive effects on stand productivity, but be mediated via structural diversity; 2. structural diversity would have a stronger effect than species diversity on stand productivity; and 3. forest stability would increase as species and structural diversity increased.

## 2. Materials and Methods

### 2.1. Study Area and Inventory Method

This study was confined to the temperate forests in South Korea (33.16° N–38.61° N, 124.59° E–130.92° E). Despite the diversity in size, these forests also had relatively diverse tree species, where on average, 12 tree species and up to 32 tree species coexisted, and were dominated by broad-leaved *Quercus* spp. and coniferous *Pinus* species. Korean National Forest Inventory data from 3959 permanent plots were used, which covered various stand ages and densities, management

intensities, and environmental conditions, including diameter at breast height (DBH), and estimated height by species–specific allometric equations for more than 700,000 trees. The NFI is a 5-year interval survey and the most recent two data periods (5th NFI during 2006–2010 and 6th NFI during 2011–2015) were used for the study.

The NFI plots consisted of a single main survey plot and three subplots located on the northern, southeastern, and southwestern sides, at a 50-m distance (Figure 1). The NFI provided regionally validated data across a wide range of spatial and temporal conditions, which monitored the long-term changes in forest stands at a large spatial scale. However, to reduce the observational errors, plots with extreme changes (>97.5th quantile) in relative stand density and basal area increments were excluded from the analysis, as the NFI did not provide full details of each site history.

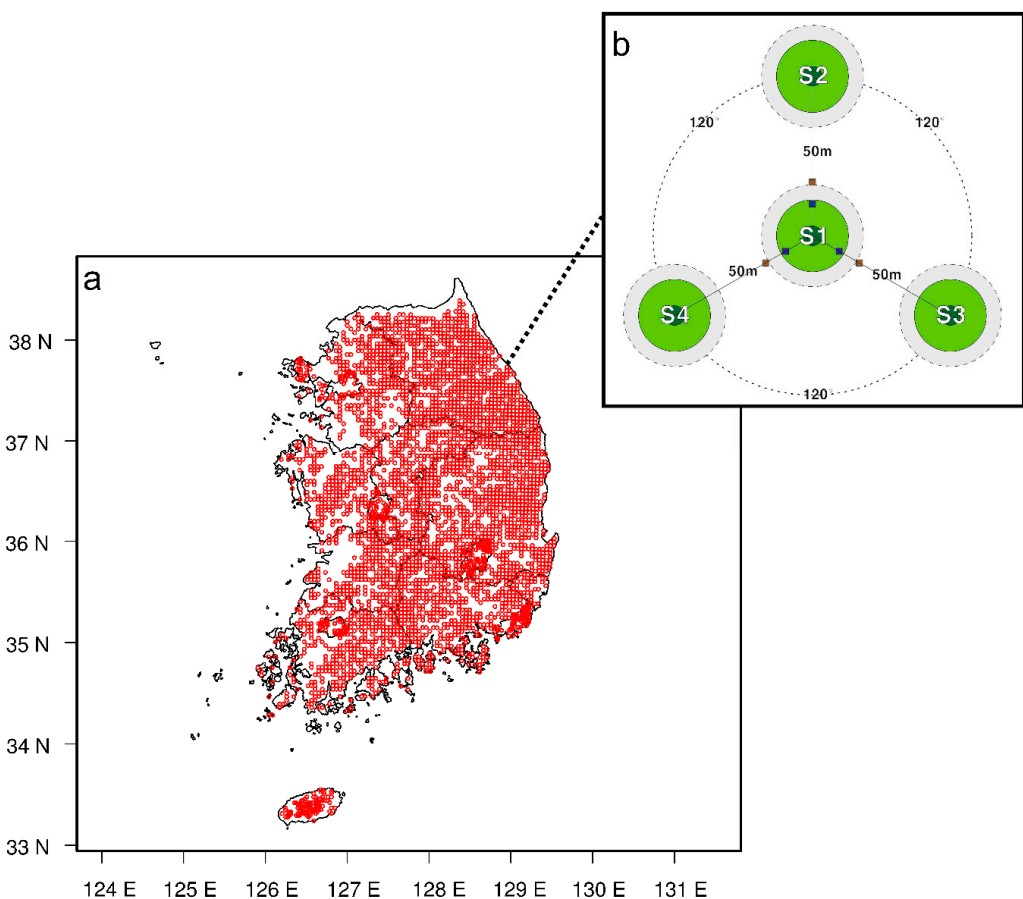

**Figure 1.** National Forest Inventory plots used in this study and their distribution across South Korea. Panel b shows the structure of the cluster plot.

## 2.2. Forest Productivity Estimation

Forest productivity was calculated at the stand level, due to limitations in NFI data for tracking individual tree growth. Stand productivity was assumed as the differences in the sum of all individual tree basal areas between the two survey periods:

$$BAI_{\text{stand}} = \sum BA6_{Tree} - \sum BA5_{Tree} \tag{1}$$

where $BAI_{\text{stand}}$ is the stand basal area increment, $BA6_{Tree}$ is the basal area of individual trees in the 6th NFI period, and $BA5_{Tree}$ is the basal area of individual trees in the 5th NFI period. The productivity analysis excluded plots with negative $BAI_{stand}$, and a total of 3151 plots were analysed.

## 2.3. Tree Species and Size Diversity

Tree species diversity ($H_{Sp}$) and structural diversity were separated for the analysis. The stand structural diversity ($H_{DBH}$) was quantified depending on variability of tree diameter. Both species and structural diversities were calculated using the Shannon diversity index [25]:

$$H_{Sp} = -\sum_{i=i}^{n} p_i \times \ln(p_i) \tag{2}$$

$$H_{DBH} = -\sum_{j=i}^{m} p_j \times \ln(p_j) \tag{3}$$

where $n$ is the total number of species in the stand, $p_i$, is the proportion of basal area for species $i$, $m$ is the total number of DBH classes in the stand, and $p_j$, is the proportion of basal area for DBH class $j$. DBH was classified in intervals of 3 cm, as this interval showed the highest correlation with stand basal area increment. Tree height and the spatial arrangement of trees was not considered in this study due to the lack of availability of height data and absence of tree coordinates. Due to high correlation among tree diameter and height (Figure S1), horizontal variability could be linked with vertical variability, which makes tree diameter variability possible to use as proxy of forest structural diversity. Other horizontal structural diversity indices, such as the coefficient of variation (CV), skewness, and closeness to a *J*-shaped distribution of DBH were not included in the analysis due to their lower correlation with productivity. Species richness (SR) was calculated using the total number of species in each plot. The stand basal area, species richness, species diversity, and structural diversity were classified into 10 groups based on their values (group intervals were 1.5 for stand basal area, 3 for species richness, and 0.3 for species and structural diversity).

## 2.4. Environmental Conditions

The environmental conditions for each plot were derived from the digital climate model (DCM). The DCM generated hyper-resolution (270 m) gridded digital reanalysis data for 30-year-averaged monthly mean air temperature and precipitation by considering landscape topographical characteristics [26,27]. Two environmental variables—annual mean air temperature and annual precipitation—were considered as main controlling factors in species and structural diversity and forest productivity. Each environmental variable was classified into three groups by quantiles, and the correlation coefficient between species and structural diversity and productivity was determined within each class.

## 2.5. Structural Equation Modelling

The effects of species and structural diversity can be separated using structural equation modelling (SEM), which identifies the interconnections and causal relationships among variables. Species diversity was assumed to have indirect effects on forest productivity via structural diversity. To test this hypothesis, we used SEM to estimate direct and indirect effects of species and structural diversity on stand BAI. The SEM in this study used a combination of factor analysis and path analysis, which did not include latent variables. This model was suitable for analysing the complex systems, while not assuming independence among variables. The causal relationships between variables were assumed to be recursive. We constructed a single SEM model that included basal area, structural diversity, and species diversity as predictors of stand BAI (Figure 2). This model represented the partial mediation of species diversity effects, so that species diversity could have both direct and indirect effects on productivity. Structural diversity was assumed to have only direct effects on productivity. We report standardized path coefficients to facilitate comparisons between pathways. The SEM was implemented using the lavaan packages [28] in *R* software.

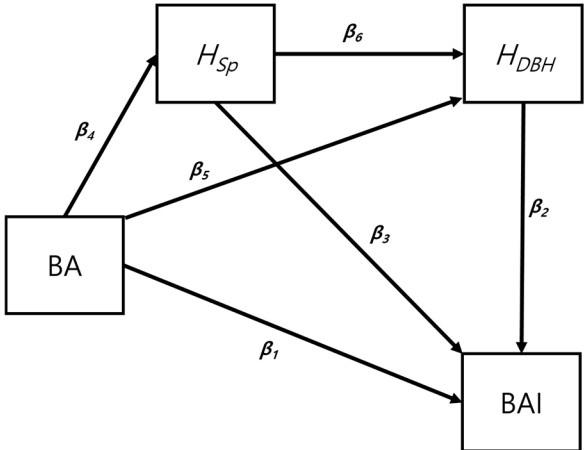

**Figure 2.** Schematic representation of the structural equation model used in the analysis. BA, $H_{Sp}$, $H_{DBH}$ and BAI stand for basal area, tree species diversity, structural diversity and basal area increment, respectively.

## 2.6. Forest Stability

In this study, forest stability was assumed to indicate consistency, and was evaluated from the structural aspect only. Because, in natural forests, changes in stand density are caused by differences in regeneration and death rate, and are usually accompanied by changes in stand structural properties like diameter distribution and species composition [29], the relative changes in stand density (ΔSD) between the two survey periods were used as a proxy of forest stability. Firstly, all the NFI data was divided into two groups depending on an increase (3238 plots) or decrease (632 plots) in stand density. In each dataset, changes in stand density were divided by stand density in the 5th NFI period, and the relative change in stand density was compared with species and structural diversity indices. Plots with drastic changes (>97.5th quantile) were considered to be affected by forest management practices, such as thinning, or natural disasters, and were excluded in the analysis. Plots with smaller changes in stand density were considered as more stable.

## 3. Results

Forest productivity was mostly affected by stand basal area, and the diversity–productivity relationship differed by index. It increased significantly with stand basal area and structural diversity, but decreased significantly with species diversity. Species richness had no effect on productivity (Figure 3). The regression lines were expressed as $BAI_{stand} = 0.0085BA_{stand} + 0.1162$, $BAI_{stand} = -0.0058H_{Sp} + 0.1575$, and $BAI_{stand} = 0.0336H_{DBH} + 0.0756$, respectively. Stand basal area was positively correlated with both species richness ($n = 3157$, $r = 0.24$, $p < 0.001$) and structural diversity ($r = 0.61$, $p < 0.001$), but was negatively correlated with species diversity ($r = -0.08$, $p < 0.0001$). Species diversity showed a positive correlation with structural diversity ($r = 0.21$, $p < 0.001$) (Figure S2).

The diversity–productivity relationships were changed by stand basal area condition. Where stand basal area was less than 3 m$^2$ per plot, both tree species and structural diversity had a positive relationship with productivity, but structural diversity had a more sensitive and significant relationship than species diversity. Under the intermediate condition, where stand basal area was higher than 3 m$^2$ per plot and lower than 6 m$^2$ per plot, the species and structural diversity had the opposite relationship with productivity. The species diversity–productivity relationship became negative as stand basal area increased. Conversely, structural diversity still had a positive effect on productivity, but the strength of the relationship was weaker than the lower stand basal area condition. Where stand basal area was higher than 6 m$^2$ per plot, the positive diversity–productivity relationship disappeared. The negative effects of species diversity on productivity increased and there was no significant relationship between structural diversity and productivity (Figure 4). The regression lines

were expressed as: $BAI_{stand} = 0.0174H_{Sp} + 0.1084$ and $BAI_{stand} = 0.0188H_{DBH} + 0.0942$, when $BA_{stand}$ was lower than 3 m$^2$; $BAI_{stand} = -0.0131H_{Sp} + 0.1733$ and $BAI_{stand} = 0.0143H_{DBH} + 0.1227$, when $BA_{stand}$ was higher than 3 m$^2$ and lower than 6 m$^2$; and $BAI_{stand} = -0.0404H_{Sp} + 0.2238$ when $BA_{stand}$ was higher than 6 m$^2$.

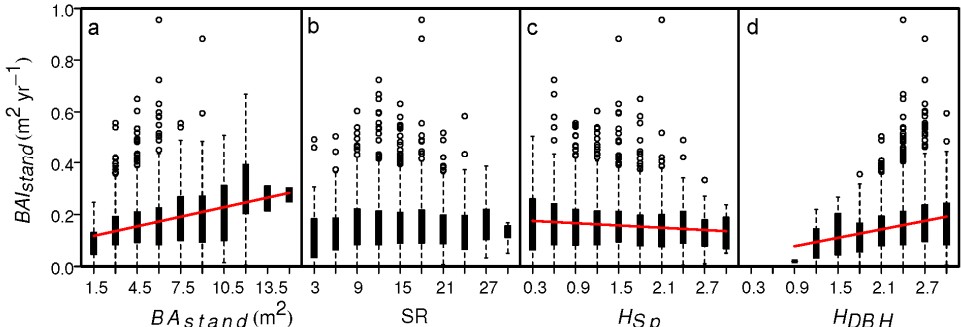

**Figure 3.** The effects of (**a**) stand basal area ($BA_{stand}$), (**b**) species richness (SR), (**c**) species diversity ($H_{Sp}$), and (**d**) structural diversity ($H_{DBH}$) on mean annual basal area increment ($BAI_{stand}$). Red lines indicate statistically significant relationships between $BAI_{stand}$ and each variable ($p < 0.001$).

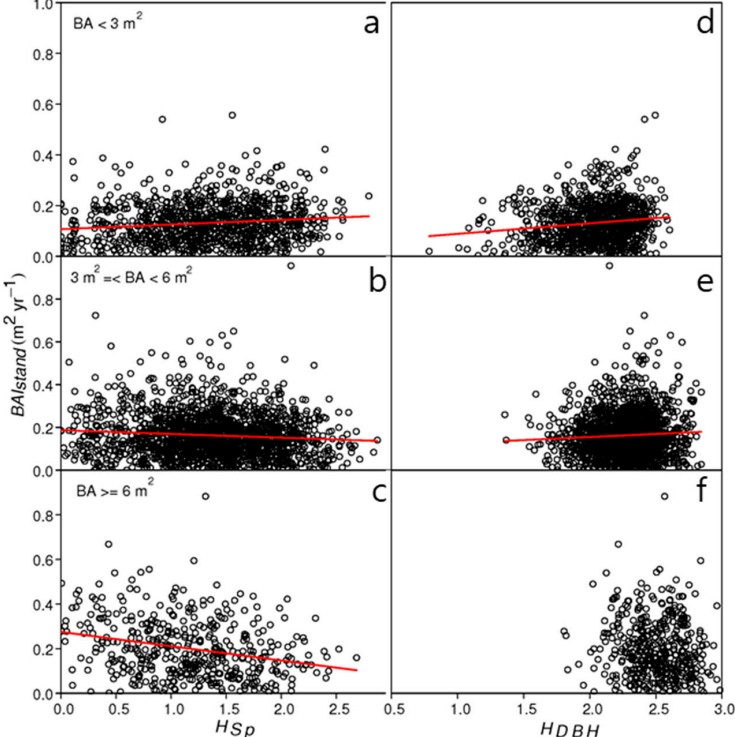

**Figure 4.** Changes in the diversity and productivity relationship by stand basal area ($BA_{stand}$). Left panels (**a**–**c**) show the relationship between stand basal increment ($BAI_{stand}$) and species diversity ($H_{Sp}$), and right panels (**d**–**f**) show the relationship between $BAI_{stand}$ and structural diversity ($H_{DBH}$) in low (top; where BA was lower than 3 m$^2$), intermediate (middle; where BA was higher than 3 m$^2$ and lower than 6 m$^2$), and high (bottom; where BA was higher than 6 m$^2$) $BA_{stand}$ plots.

The slope between productivity and diversity indices was changed by environmental conditions (Figure 5). The positive relationship between productivity and structural diversity was strongest on plots under high annual mean temperature conditions, but was highest at intermediate precipitation conditions. Conversely, the negative relationship between productivity and species diversity was strongest on low air temperature conditions, but the trend was opposite with precipitation.

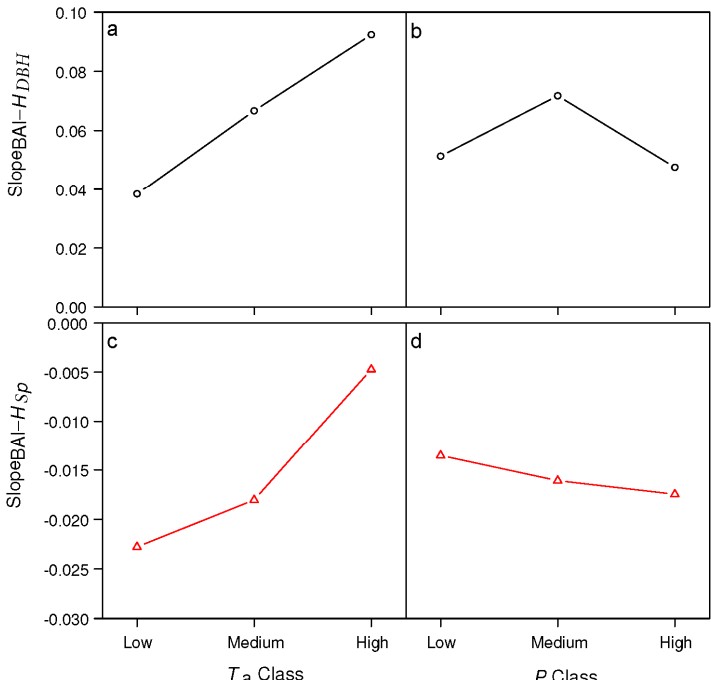

**Figure 5.** Changes in slope between (**a**) productivity and structural diversity ($H_{DBH}$; top) by annual mean air temperature, and (**b**) productivity and species diversity by annual mean air temperature ($H_{Sp}$; bottom), (**c**) productivity and structural diversity by annual precipitation, and (**d**) productivity and species diversity by annual precipitation. Red indicates negative values.

Figure 6 shows that stand basal area and structural diversity had direct positive effects on stand productivity ($\beta$ = 0.26 and 0.09, respectively, both $p < 0.001$), but species diversity showed a negative direct effect ($\beta$ = −0.09, $p < 0.01$). Stand basal area also had a direct effect on structural diversity ($\beta$ = 0.61, $p < 0.01$), but a negative effect on species diversity. The correlation between basal area and species diversity was negative and the opposite correlation was found in structural diversity (Figure S2), where it had a positive indirect influence on stand productivity. Species diversity had a positive indirect effect via structural diversity ($\beta$ = 0.26, $p < 0.001$).

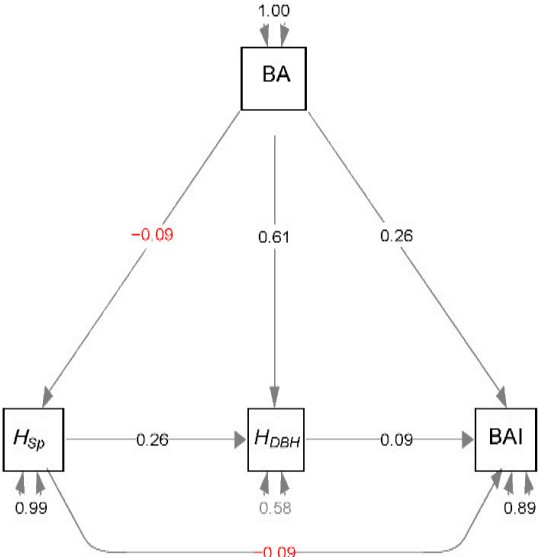

**Figure 6.** Structural equation model relating stand basal area increment ($BAI_{stand}$) to stand basal area ($BA_{stand}$), species diversity ($H_{Sp}$), and structural diversity ($H_{DBH}$). Solid arrows represent significant ($p < 0.05$) paths and standardized regression coefficients are shown. The red numbers indicate the negative effects.

Structural diversity had a significantly positive effect on forest stability, but species diversity showed no such relationship (Figure 7). As structural diversity increased, the relative increment of stand density decreased and the relative decrement of stand density also decreased. The regression lines were expressed as: $\Delta SD = -0.0376 H_{DBH} + 0.4721$ in plots with positive $\Delta SD$, and $\Delta SD = 0.0273 H_{DBH} - 0.4121$ in plots with negative $\Delta SD$. This resulted in less temporal variation in stand density (between the two survey periods) on more structurally diverse plots. Conversely, species diversity did not have an impact either on relative increment of stand density, or on relative decrement of stand density.

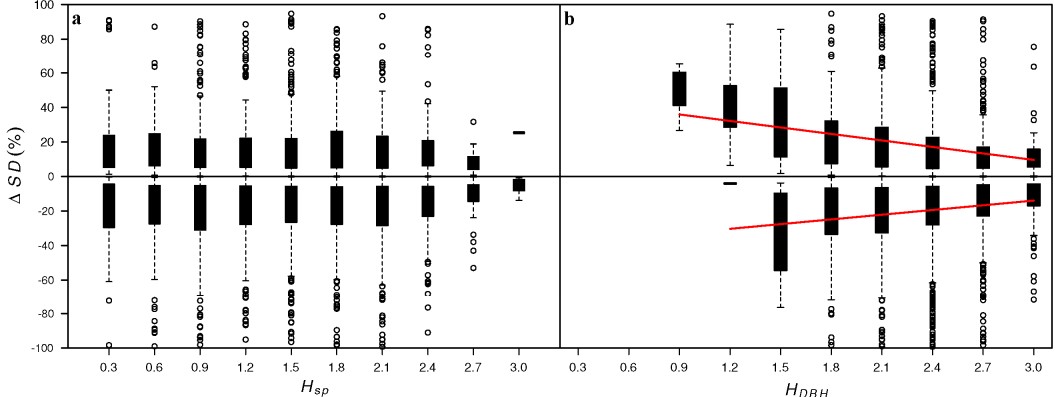

**Figure 7.** Changes in relative stand density ($\Delta SD$) by (**a**) species diversity ($H_{Sp}$) and (**b**) structural diversity ($H_{DBH}$). Red lines indicate linear regression lines between $H_{DBH}$ and $\Delta SD$ ($p < 0.001$).

## 4. Discussion

The results of this study showed that structural diversity had a stronger effect than species diversity on forest productivity. Similarly, in mixed, uneven-aged forests in Germany, structural diversity had a profound effect on forest stand productivity, whereas species diversity effects were partially mediated via structural diversity [16]. The positive interaction between structural diversity and productivity was also found in a spruce-dominated forest in Canada [20,30]. Conversely, structural diversity has been shown to have a negative effect on productivity in coniferous forests [31] and also in deciduous forests [19,32]. Insignificant interactions in stand structure and productivity were also reported [33]. The variation in interactions between structural diversity and productivity may partly be caused by differences in the methods used to estimate the degree of structural diversity, but are mainly caused by complex interactions among species diversity, structural diversity, and stand characteristics. For example, structural diversity caused up to a 13% reduction in stand productivity via management techniques [34], which induced less competition for water and nutrients and low light use efficiency of small trees under limited interaction with neighboring trees.

The positive effects of structural diversity on stand productivity could have been caused by a high degree of canopy complexity, which is a potential driver of forest productivity [35,36]. Due to the significant correlation between tree diameter and height (Figure S1), the plots with higher structural diversity in this study might have more diverse height distribution. A diverse canopy structure enables more light penetration to understory species, which maintains greater stand total leaf area and increases stand productivity. Canopy complexity can also enhance photosynthetic efficiency. When leaf area driven increases in stand productivity reach a plateau, stand productivity continuously increases with canopy roughness [37]. Canopy complexity and structural diversity can occur through plasticity, when species adjust crown allometry to maximize light interception [38,39].

In temperate forests, species diversity may control productivity differently via structural diversity. There are many underlying processes that are related to stand structural and species compositional attributes. Enhanced productivity via nitrogen fixation is correlated more with species composition, which is more related to species identity, such as nitrogen fixing capacity, than to species diversity. A positive interaction between complementarity and stand density [40] is more sensitive to structural

than species diversity. The relative contributions of two different types of diversity to reducing competition in productivity determines the direction of diversity-productivity interactions [41]. The most productive forests include mono species *Eucalyptus* plantations, which are managed as monocultures to increase productivity. Similarly, the maximum productivity was decreased with species richness in European temperate forests [42]. In Korea, a long history of reforestation has resulted in nation-wide distribution of monocultures and even-aged forests. These include middle-aged *Larix* forests, which have higher productivity than mixed forests. Species diversity cannot improve productivity in a stand that is near its optimum efficiency. Forest management practices, such as pruning and fertilization, lessen competition, and reduce the sensitivity to species diversity. Otherwise, species diversity could be increased under poor site conditions, where open canopy conditions enable the mixture of small tree species with lower productivity [43]. This would also cause a negative correlation between species diversity and stand basal area. In addition, it is possible that the species diversity index did not capture the entire range of shade tolerances nor account for the effect of vertical crown stratification. The structural diversity index could have better reflected the ability to quickly and efficiently fill new gaps more than species diversity.

Stand basal area can significantly affect the diversity and productivity relationship. When stand basal area is low, any interactions among trees are weak, and intensify as stand basal area increases; however, the interaction direction can change with site conditions. Changes in direction of interactions between diversity and productivity are determined by the relative strength of the complementarity effect [44] and competition intensity [45,46], with increasing stand density or basal area. The diversity-productivity relationship can also be changed depending on the environmental conditions. In this study, the slope between structural or species diversity and productivity changed by air temperature and precipitation conditions, and the changes by air temperature were more significant than precipitation. Although air temperature had little influence on productivity, it negatively affected both species and structural diversity. Conversely, precipitation has a positive relationship with productivity, species, and structural diversity (Table S1).

The stability in stand density was increased with structural diversity, but was not affected by species diversity. Even though stability in stand density does not represent the whole stability of a forest stand, the results contradict previous studies that reported a positive association between species diversity and stability [47,48]. The positive relationship is mainly dependent on asynchronous species responses to environmental fluctuations and an increased possibility of over-yielding by species mixture [49]. The higher stability of stand density in structurally diverse forests could cause increased productivity. The long-term productivity of a forest is closely linked to its internal nutrient cycling [50,51], with structurally stable forests having higher buffer capacity [52]. Because buffer capacity includes storage capacity for water and nutrients, and resistance to acidification and nutrient leaching, structurally diverse forests have increased biogeochemical stability, which would lead to higher productivity under various stress conditions.

## 5. Conclusions

Forest structural diversity promoted both productivity and stability in stand density of temperate forests. Although species diversity had an overall negative effect on productivity and a neutral effect on stability in stand density, it had an indirect positive effect on productivity via structural diversity. Due to the large heterogeneity of the stand condition and sensitivity of the diversity–productivity relationship on climatic conditions, these results do not confirm the general rule of higher productivity in less species-diverse forests. Rather, they indicate that in temperate forests, structural diversity is a key contributor of forest sustainability by promoting productivity and stability in stand density. Therefore, forest management planning should focus on creating structurally diverse forests.

**Supplementary Materials:** The following are available online at http://www.mdpi.com/1999-4907/10/12/1113/s1, Figure S1: The relationship between tree diameter at breast height (DBH) and measured tree height from 5th National Forest Inventory data ($r = 0.65$, $p < 0.0001$), Figure S2: Correlation plots among stand density (SD), stand

basal area (BA), species richness (SR), species diversity ($H_{Sp}$), structural diversity ($H_{DBH}$), and basal area increment (BAI). Numbers indicates Pearson's correlation coefficients and asterisks indicate statistical significance (*$p < 0.05$, ** $p < 0.01$, *** $p < 0.001$), Table S1: Correlation table between environmental conditions (air temperature ($T_A$), and precipitation ($p$)) and stand characteristics (species diversity ($H_{Sp}$), structural diversity ($H_{DBH}$), and basal area increment (BAI)).

**Author Contributions:** Conceptualization: J.P.; Formal Analysis: J.P.; Data curation: H.K.J., I.B.J.; Writing–original draft: J.P. and H.S.K.; Visualization: J.P.; Supervision: H.S.K.; Project Administration: H.S.K.; Funding acquisition: H.S.K.

**Funding:** This research was supported by Korea Forest Service (S211315L020120, S111215L020110), Promising-Pioneering Researcher Program through Seoul National University (SNU) in 2015, and Seoul National University Big Data Institute through the Data Science Research Project 2016.

**Acknowledgments:** We would like to acknowledge the extensive efforts of NFI field crews and staff for collecting, curating, and distribute the field measurement data used in this study.

**Conflicts of Interest:** The authors declare no conflict of interest.

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
