# Peer review of "The Influence of Tree Structural and Species Diversity on Temperate Forest Productivity and Stability in Korea"

_forests, doi:10.3390/f10121113_

Round 1
Reviewer 1 Report
Structural, Not Species Diversity, Enhances Temperate Forest Productivity and Stability
The paper presents an elaboration of National Forest Inventory (NFI) data collected in South Korea of ten (2006-2010) years period at two 5-year intervals.
The basic parameters of stand productivity were calculated e.g. the stand basal area increment, the basal area of individual trees, Species diversity (HSp), structural diversity (HDBH) and Shannon diversity index. The environmental conditions for each plot were derived from digital climate model (DCM). DCM generated hyper-resolution (270 m) gridded digital reanalysis data for 30-year-averaged monthly mean air temperature and precipitation by considering landscape topographical characteristics.
The following hypothesizes were stated;
Species diversity would have positive effects on stand productivity, but be mediated via structural diversity; Structural diversity would have a stronger
effect than species diversity on stand productivity; Forest stability would increase as species
81 and structural diversity increased.
The introduction possesses a very general character typical to rather review paper, not the original one.
Methods of elaboration were presented correctly. The methods could be improved by using a little bit more precise indexes involving e.g. the quality of forest type soil, fertility, ecological indicator values, detailed analysis species frequency (species richness). The general level of scientific analysis does not possess too high-value scientific impact.
The manuscript could be improved by more detailed characteristic analyses of available data-set. The data-set possesses great potential to perform many interesting studies. The productivity of the ecosystem, as the value/quantity of production and its pace in a given ecosystem over a given period of time, can be used to construct an indicator of the diversity of this ecological system and comparisons with other ecosystems.
The way of describing the results with a lot of numerical data is difficult for the reader to follow the trends of the research results. The conclusions have a very laconic character of the description.
Overall merit the elaboration presented in the manuscript is good background to further analysis.
Reviewer 2 Report
The manuscript is based on the analyses of a large data set and this is very significant ad valuable!
Nevertheless, I have the feeling that the data set is so heterogeneous and includes a variety of ecological conditions and managements practices (planted monocultures, fertilization) that finally the results and discussion presented in this paper is an oversimplification and over generalization.
Discussion is not sound. There is a limited discussion of the results and authors fail to bring their results within a conceptual framework. The manuscript is overambitious and there are conceptual gaps between their (interesting) results and the discussion of these results. I strongly recommend authors to discuss their findings within the limits of their data and the limits of the questions that are addressed by their analysis. For example there is no question regarding the effect of environmental conditions in forest productivity and forest stability is ill-defined (for both example see specific comments bellow). Additionally, authors should discuss their findings through their comparison with other similar studies and demonstrate how their results are supported from similar studies and how (and why) there is controversy with studies resulting is different outcomes.
Moreover, there are some issues that should be addressed by the authors in order to clarify their approach.
Material & methods, l. 108-123: I fell like the title is misleading. The term “Species diversity” should be “canopy species diversity” or “tree species diversity”. Species diversity refers to the entire plant community, while authors refer to the diversity of canopy trees. The term “Structural diversity” should be replaced by “diameter diversity” or something similar. Structural diversity includes far more parameters than just the DBH. Material & methods, l. 122-123: I see some interesting but surprising figures: 10 groups and groups intervals by 3 for species richness implies that there can be plots with than 30 tree (canopy) species (see also figure 2). This is an interestingly large number for temperate oak or pine forests. The authors should present some extreme cases like this. Material & methods, l. 124-131 and Discussion l. 250-255: Although this part is not of “core importance”, I do not believe that annual values of air temperature or precipitation totals can describe the environmental conditions of temperate forests. Air temperature and precipitation totals during the growth period are more informative and NFI pots should be separated on the basis of these environmental conditions. Discussion should also be focused on the effect of these environmental variables and not on annual values. Material & methods, l. 124-131: If the authors developed the hyper-resolution gridded data of the climatic variables, they should present the methods they followed. Otherwise they should cite their sources. Material & methods, l. 125: Authors should cite the source of the DCM model they used. Material & methods, l. 132-145: Authors are encouraged to schematically demonstrate their model’s structure (similarly to figure 5). Material & methods, l. 146-152: Authors should demonstrate and clarify to readers that there are not exogenous parameters influencing forest stability variables. Stand density can be modified by management practices and this may have an important impact in the outputs. Authors should demonstrate how they excluded NFI plots in which clearings or any other above-ground biomass removal due to management of natural causes (e.g. wind and snow breaks) was observed. Material & methods, l. 146-152: I can not agree with the concept of authors that stability can be represented by change in stand density. On the introduction and on the discussion they shortly present some (correct) aspects of stability that include resistance and resilience. Change in stand density can not represent neither resistance nor resilience. Authors should not mention that their study addresses forest stability and simply directly state the they examine the relationship between relative stand density and the explanatory variables they are testing. Results l. 179-183 & Discussion l. 250-255: I do not thing that the authors actually tested the effect of environmental conditions on the slope between productivity and diversity indices. They rather examined how the slope of the relationship changes in different air temperature and precipitation classes. Results, l. 191-192: Authors should explain and discuss (directly and clearly) the negative effect of stand basal area to species diversity. Discussion l. 222-223: Authors should discuss how do they relate structural diversity (they estimate it though diversity in DBH) and canopy complexity. They never present such an analysis in order to demonstrate this relationship, nor they cite any literature that demonstrates a direct relationship between DBH and canopy complexity. Discussion l. 229-244: I see that the authors refer to plantations and fertilization. It could be interesting to perform the analyses separately for forest plantations (artificial monocultures) and (semi)natural forests. These two categories differ significantly ecologically and conceptually. Similarly, NFI plots in stands that fertilization takes place should be excluded or treated separately. I feel that the data-set although large, it is very heterogeneous and can not ensure sound outputs. Discussion l. 231-232: Authors differentiate species composition from species diversity. They should define the term species composition. Discussion l. 250-255: I do not thing that the authors actually tested the effect of environmental conditions on the slope between productivity and diversity indices. They rather examined how the slope of the relationship changes in different air temperature and precipitation classes.
Round 2
Reviewer 1 Report
The title of the manuscript -Tree Size, Not Species Diversity, Enhances Temperate Forest Productivity
is not any kind of general title but a hypothesis I would suggest some change e.g. The influence of tree size and species diversity on Temperate Forets Productivity in South Korea.
Response of Authors
Point 3: The manuscript could be improved by more detailed characteristic analyses of available data set. The data-set possesses great potential to perform many interesting studies. The productivity of the ecosystem, as the value/quantity of production and its pace in a given ecosystem over a given
period of time, can be used to construct an indicator of the diversity of this ecological system and comparisons with other ecosystems.
Response 3: Similar to response 2, 7th NFI survey will finished in next year. After that we will analyse the changes of productivity and main cause of the change.
I could not see too many changes between first and improved versions of manuscripts.
Author Response
Point 1: The title of the manuscript -Tree Size, Not Species Diversity, Enhances Temperate Forest Productivity is not any kind of general title but a hypothesis I would suggest some change e.g. The influence of tree size and species diversity on Temperate Forest Productivity in South Korea.
Response 1: We changed the title as you suggested(L1-2)
Point 2: Point 3: The manuscript could be improved by more detailed characteristic analyses of available data set. The data-set possesses great potential to perform many interesting studies. The productivity of the ecosystem, as the value/quantity of production and its pace in a given ecosystem over a given period of time, can be used to construct an indicator of the diversity of this ecological system and comparisons with other ecosystems.
Response 3: Similar to response 2, 7th NFI survey will finished in next year. After that we will analyse the changes of productivity and main cause of the change.
I could not see too many changes between first and improved versions of manuscripts
Response 2: We are testing the various attributes for explaining the biases in productivity-diversity relationships. Unfortunately, those analyses could not be included in this paper due to time limitations in paper processing. We will report those results in the next paper.
Reviewer 2 Report
The manuscript is adequately improved.
Author Response
Thank you for the kind comment.